# Scattering Amplitude of Surface Plasmon Polariton Excited by a Finite Grating

**DOI:** 10.3390/nano13142091

**Published:** 2023-07-17

**Authors:** Anton V. Dyshlyuk, Alexey Proskurin, Andrey A. Bogdanov, Oleg B. Vitrik

**Affiliations:** 1Institute of Automation and Control Processes, Far Eastern Branch of the Russian Academy of Sciences, Vladivostok 690041, Russia; anton_dys@mail.ru; 2School of Engineering, Far Eastern Federal University, Vladivostok 690090, Russia; 3School of Information Technologies, Vladivostok State University, Vladivostok 690014, Russia; 4School of Physics and Engineering, ITMO University, St. Petersburg 197101, Russia; alexey.proskurin@metalab.ifmo.ru (A.P.); a.bogdanov@metalab.ifmo.ru (A.A.B.); 5Qingdao Innovation and Development Base, Harbin Engineering University, Sansha Road 1777, Qingdao 266000, China

**Keywords:** surface plasmon polaritons, SPP, SPP excitation, nanostructured substrates

## Abstract

Unusual optical properties of laser-ablated metal surfaces arise from the excitation of local plasmon resonances in nano- and microstructures produced by laser-processing and from the mutual interaction of those structures through surface plasmon polariton (SPP) waves. This interaction provides a synergistic effect, which can make the optical properties of the composite nanostructure drastically different from the properties of its elements. At the same time, the prediction and analysis of these properties are hampered by the complexity of the analytical solution to the problem of SPP excitation by surface objects of arbitrary configuration. Such a problem can be reduced to a simpler one if one considers the geometry of a structured surface as a superposition of harmonic Fourier components. Therefore, the analytical solution to the problem of surface plasmon polariton excitation through the scattering of light by a sinusoidally perturbed plasmonic metal/vacuum boundary becomes very important. In this work, we show that this problem can be solved using a well-known method for calculating guided-mode amplitudes in the presence of current sources, which is used widely in the waveguide theory. The calculations are carried out for the simplest 2D cases of (1) a sinusoidal current of finite length and (2) a finite-length sinusoidal corrugation on a plasmonic metal surface illuminated by a normally incident plane wave. The analytical solution is compared with the results of numerical simulations. It is shown that, in the first case, the analytical and numerical solutions agree almost perfectly. In the second case, the analytical solution correctly predicts the optimum height of the corrugation $x_{\mathrm{opt}}$, providing the maximum SPP excitation efficiency. At the same time, the analytical and numerical values of the SPP amplitude agree very well when the corrugation height *x* turns out to be $x\ll x_{\mathrm{opt}}$ or $x\gg x_{\mathrm{opt}}$ (at least up to $3x_{\mathrm{opt}}$); at $x=x_{\mathrm{opt}}$, the mismatch of those does not exceed 25%. The limitations of the analytical model leading to such a mismatch are discussed. We believe that the presented approach is useful for modeling various phenomena associated with SPP excitation.

## 1. Introduction

Nanomaterials require a specialized toolkit to construct on-chip electronics and develop highly efficient devices capable of concentrating light at the nanoscale. To overcome the limitations of traditional dielectric waveguide nanotechnology, one promising approach involves the utilization of metallic structures that support plasmon modes. The field of plasmonics offers a compelling solution by harnessing the collective oscillations of electrons at the surface of these metallic nanostructures. This allows for intense light–matter interactions and enables the precise manipulation and control of light [1].

Nowadays, the field of plasmonics is a rapidly growing area of research that has gained significant attention in recent years due to its potential applications in various fields, including biosensing [2,3,4], fiber sensing [5,6], imaging [7,8,9], data storage [10], and energy conversion [11,12]. The field of plasmonics focuses on the interaction between electromagnetic radiation and free electrons in metal structures, known as surface plasmon polaritons (SPPs), which can lead to unusual optical properties and strong light–matter interaction.

The history of plasmonics dates back to the early 20th century, when SPPs were first discovered as anomalies in the reflectance spectra [13]. However, it was not until the 1950s that researchers started to study the interaction between light and metal surfaces and realized the potential of surface plasmons for enhancing the absorption and scattering of light. Since then, the field of plasmonics has evolved rapidly, with numerous studies exploring the properties and applications of SPPs in various structures, including metallic nanoparticles [14], thin films [15], and nanostructures created by laser ablation [16,17].

Optical phenomena involving SPPs can occur at subwavelength scales and are not limited by the diffraction limit. Therefore, one of the most promising applications of plasmonics is in the field of nanophotonics, which involves the manipulation of light at the nanoscale. This field has been driven by the need to develop smaller and more efficient optical components for a wide range of applications, from telecommunications [18] to medical imaging and drug delivery [19]. Plasmonic coupling between nanoantennas over a conductive surface is the basis for the design of highly sensitive sensors [20] and thermoplasmonic devices [21].

However, the design and optimization of plasmonic nanostructures for specific applications can be challenging, as it requires a deep understanding of the complex interactions between light and matter at the nanoscale. The excitation of SPPs in these structures is particularly important, as it can enhance the local electromagnetic field and lead to increased light–matter interactions. Therefore, there is a need for accurate and efficient computational tools for modeling the behavior of SPPs in plasmonic nanostructures, which can aid in the design and optimization of these structures for various applications.

The study and analysis of nanostructured plasmonic surfaces raise even broader questions. In addition to the excitation of SPPs in individual structures, the mutual interactions between neighboring structures can also lead to novel optical effects. This interaction arises from the coupling of SPPs between adjacent structures, which can result in the formation of new modes with different spectral and spatial properties. The resulting composite structures exhibit a synergistic effect that is not present in the individual structures, leading to significant changes in the optical properties.

In this work, we extend the approach initially suggested by us in [22,23] for a description of SPP excitation by confined nanoantennas to the case of SPP excitation by nanostructured substrates. While modern manufacturing techniques, including laser ablation, allow the creation of arbitrarily shaped profiles, we investigate the simplest case of a sinusoidally perturbed surface because the general scattering problem can be decomposed into the set of Fourier components. We pay extra attention to properly accounting for SPP radiation losses and the impact of the finite length of the profiled area. All the analytical results are confirmed by the numerical calculations.

## 2. Results and Discussion

### 2.1. Excitation of SPP within Born Approximation

First, we are going to determine the efficiency of the induced SPP wave based on the Lorentz reciprocity theorem [24]. Suppose we have a finite-length grating on the metal–air interface under the plane wave illumination (Figure 1a).

To calculate the amplitude of the excited plasmon polaritons, we use the expression following the unconjugated form of the Lorentz reciprocity theorem [24]
(1)$$a_{\mathrm{SPP}}=-\frac{1}{4N_{0}}\int_{\left(V\right)}\left(e_{\mathrm{SPP}}\cdot J\right)\;e^{-ik_{\mathrm{SPP}}z}dV,$$
where $e_{\mathrm{SPP}}=e_{x}^{\mathrm{SPP}}n_{x}+e_{z}^{\mathrm{SPP}}n_{z}$ is the SPP mode’s electric field with $e_{z}^{\mathrm{SPP}}\left(x\right)=e_{0}e^{-\gamma \left(x\right)\left|x\right|}$ and $e_{x}^{\mathrm{SPP}}\left(x\right)=e_{0}\frac{\sqrt{\varepsilon_{\mathrm{Me}}}}{\varepsilon_{S}\left(x\right)}e^{-\gamma \left(x\right)\left|x\right|}$; $e_{0}=-h_{0}\frac{\rho_{V}}{\sqrt{\varepsilon_{\mathrm{Me}}+1}}$; $\rho_{V}=\sqrt{\frac{\mu_{0}}{\varepsilon_{0}}}$; $h_{\mathrm{SPP}}=h_{y}^{\mathrm{SPP}}n_{y}$ is the SPP mode’s magnetic field with $h_{y}^{\mathrm{SPP}}=h_{0}e^{-\gamma \left(x\right)\left|x\right|}$; $\varepsilon_{S}\left(x\right)=\left\{\begin{array}{cc} 1, & x⩾\;0\\ \varepsilon_{\mathrm{Me}}, & x<0 \end{array}\right.$; $\gamma \left(x\right)=\left\{\begin{array}{cc} \gamma_{\mathrm{Vac}}, & x⩾\;0\\ \gamma_{\mathrm{Me}}, & x<0 \end{array}\right.$; $h_{0}$ is the arbitrary constant of dimension A/m; $N_{0}=\frac{1}{2}\left|\int_{\left(S\right)}e_{\mathrm{SPP}}\times h_{\mathrm{SPP}}\cdot n_{z}\;dS\right|$; $\gamma_{\mathrm{Vac}}=k_{0}\frac{i}{\sqrt{\varepsilon_{\mathrm{Me}}+1}}$; $\gamma_{\mathrm{Me}}=-k_{0}\frac{i\varepsilon_{\mathrm{Me}}}{\sqrt{\varepsilon_{\mathrm{Me}}+1}}$; $k_{\mathrm{SPP}}=k_{0}n_{\mathrm{SPP}}$; $n_{\mathrm{SPP}}=\frac{\sqrt{\varepsilon_{\mathrm{Me}}}}{\sqrt{\varepsilon_{\mathrm{Me}}+1}}$; $n_{x}$, $n_{y}$ are the unit vectors in the *X* and *Y* directions, respectively; J is the excitation current density distribution; and $k_{0}=2\pi /\lambda$ is the vacuum wavenumber.

Let us assume that this grating represents a corrugation with periodically varying height $x_{g}=x_{0}\cos k_{g}z$ and total length $z_{0}$ (Figure 1a), and the height of the corrugation $x_{0}$ is small compared to its period $\Lambda =2\pi /k_{g}$. Suppose that a plane electromagnetic wave with a wavelength λ not too different from the period Λ falls normally on this corrugation from the vacuum (see Figure 1a). The density of the equivalent current, due to the presence of the corrugation, can be described as in [24]:(2)$$J_{g}=-ik_{0}\left(\varepsilon \left(x,z\right)-\varepsilon_{s}\left(x\right)\right)\;\frac{E}{\rho_{V}},$$
where E is the electric field amplitude, the dependencies ε(x,z) and $\varepsilon_{s}\left(x\right)$ set the modulation of the dielectric permittivity of the layer of the considered plasmonic metal with corrugated and initial flat surfaces, respectively. The field E is a superposition of the incident wave and that reflected from the metal surface with some additional contribution associated with light scattering.

While the E itself depends on the SPP field and the latter is described by the $a_{\mathrm{SPP}}$ amplitude, (Equation 1) defines a self-consistent problem. In order to simplify the problem, we consider using the first Born approximation [25]. For this purpose, we will ignore all the scattering terms in (Equation 2), and assume that
(3)$$E=E_{0\_\mathrm{ins}}n_{z}\left\{\begin{array}{cc} e^{ik_{0}x}+re^{-ik_{0}x}, & x⩾0\\ \left(1+r\right)e^{\gamma_{\mathrm{Me}}^{0}x}, & x<0 \end{array}\right.,$$
where $E_{0\_\mathrm{ins}}$ is the incident wave field amplitude, $r=\left(1-\sqrt{\varepsilon_{\mathrm{Me}}}\right)/\left(1+\sqrt{\varepsilon_{\mathrm{Me}}}\right)$ the reflection coefficient from the flat metal surface, and $\gamma_{\mathrm{Me}}^{0}=ik_{0}\sqrt{\varepsilon_{\mathrm{Me}}}$.

Once J is set, the integral (Equation 1) can be evaluated analytically, provided that $x_{0}\ll \lambda$ by a power decomposition of a small parameter $k_{0}x_{0}$. After taking into account the linear terms only, one can obtain the following expression for the *Z* component of the electric field right after the end of the grating:(4)$$E_{0z}=\frac{1}{\sqrt{1+\varepsilon_{\mathrm{Me}}}}\frac{k_{\mathrm{SPP}}^{3}k_{g}}{k_{0}\left(k_{\mathrm{SPP}}^{2}-k_{g}^{2}\right)}\left(1+r\right)x_{0}\left(1-e^{\left(-k_{I}+i\Delta k\right)z_{0}}\right)E_{0\_\mathrm{ins}},$$
where $k_{\mathrm{SPP}}=k_{R}+ik_{I}$, kR=RekSPP, kI=ImkSPP, $\Delta k=k_{R}-k_{g}$.

One can find a derivation of (Equation 4) for the case when the total length of the grating possesses the form $z_{0}=\left(n+\frac{1}{2}\right)\Lambda$ (where *n* is an integer) from the Lorentz reciprocity theorem in Appendix A.

To verify the analytical results obtained, we performed numerical simulations using the commercial software COMSOL Multiphysics. In this subsection, we create a two-dimensional numerical model simulating half of a grating with a total length of (2n+1/2)Λ (with the perfect electric conductor boundary condition to ensure mirror symmetry; *n* is considered an integer) surrounded by a perfectly matched layer. The solution to the Maxwell equation was obtained using the finite-element method in the frequency domain (FEFD). The SPP contribution was determined from Equation (Equation 1) with the calculated electric field distribution substituted into (Equation 2). Integration in Equation (Equation 1) was performed along a vertical line perpendicular to the interface, positioned after the sinusoidal perturbation. We chose the simulation area size and mesh size to ensure that the results did not change significantly under small variations of the model’s parameters.

Figure 2 compares the analytical calculations according to (Equation 4) and the results of full-wave numerical calculations for different grating periods. Hereinafter, we consider gold as a plasmonic metal (dielectric permittivity of gold is taken from [26]). The figure demonstrates that, overall, (Equation 4) agrees with the results of the simulations for relatively low gratings (namely, for grating heights up to 30 nm). As the length of the grating increases, the distinction of the dependencies grows stronger. Moreover, the degree of divergence depends on the grating period. When the difference between the resonant period and the actual one (red color in Figure 2; the period equals 778 nm) suppresses the SPP generation, the analytical expression describes the behavior of the numerical results even for longer grating lengths of up to 300 μm. However, while the grating period approaches the resonance, the divergence becomes more pronounced. One can explain it as follows. The Born approximation assumes that the background field largely determines the total electric field value. If this condition is met, the additional terms related to the “self-action“ of the SPP field are negligible In the case of resonant excitation, the SPP field is compatible in strength with the background field; therefore, the approximation is less accurate.

Another factor limiting the accuracy of the described approach is the grating amplitude. Indeed, the background field decays as it penetrates the metal. By contrast, the total electric field is essentially non-zero near the grating, even for the larger values of $x_{0}$. Thus, the background field is no longer the dominant contribution to the total field.

Figure 3 supports this conclusion. Indeed, (Equation 4) adequately describes the numerical experiment for small grating heights (up to 5 nm). A further increase in the grating amplitude leads to a divergence of the numerical results from the analytically predicted ones. Importantly, the difference is retained even when the next orders of magnitude of the parameter $k_{0}x_{0}$ are taken into account, despite the corresponding expressions becoming more complicated.

Therefore, there is a need for another approach, which will better describe the SPP generation for larger grating amplitudes, as well as the case of resonant SPP excitation.

### 2.2. Excitation of SPP by a Flat Harmonic Current

Now, we will explore a slightly different approach. Firstly, let us consider the problem of plasmon excitation by a harmonic current distributed on a flat metal–dielectric interface. The current density possesses the form $J_{\mathrm{surf}}=J_{\mathrm{surf}}\cdot n_{z}$, where $n_{z}$ is a unit vector in the *Z*-axis direction,
(5)$$J_{\mathrm{surf}}=\delta \left(x\right)I_{\mathrm{surf}}\cos k_{g}z,$$
δ(x) is the Dirac delta function, $k_{g}=\frac{2\pi }{\Lambda }$, $I_{\mathrm{surf}}$ is the current amplitude, which implicitly includes the time dependence exp(iωt), Λ is the current modulation period along the *Z*-axis (Figure 1b). The current is not infinite in this direction, and its range is limited by the length $z_{0}$.

Assuming in (Equation 1) that $J=J_{\mathrm{surf}}$, we note that of the two terms, $\exp(ik_{g}z)$ and $\exp(-ik_{g}z)$, which are in the Euler representation for the function $\cos k_{g}z$, defining the spatial modulation of the excitation current density in relation (Equation 5), only the former one makes a significant contribution to the amplitude of the forward-propagating (i.e., in the *Z* axis direction) SPP, while the contribution of the second one is negligibly small. The situation is the opposite for the back-propagating plasmon. Taking into account only the main contribution to the amplitude of the forward-propagating SPP, the *Z* component of its electric field $E_{z}=a_{\mathrm{SPP}}e_{z}^{\mathrm{SPP}}$ can be written as $E_{z}=E_{0z}e^{-\gamma (x)|x|}$, where
(6)$$E_{0z}=i\gamma_{I}\frac{1}{\Delta k+ik_{I}}\left(1-e^{-z_{0}k_{I}+i\Delta kz_{0}}\right)\rho_{V}I_{\mathrm{surf}},$$
and $\gamma_{I}=\frac{k_{0}n_{\mathrm{SPP}}^{2}}{2\varepsilon_{\mathrm{Me}}}\left|\frac{\varepsilon_{\mathrm{Me}}^{3/2}}{\varepsilon_{\mathrm{Me}}^{2}-1}\right|$.

The results of calculations for the amplitude of this component, as carried out by expression (Equation 6) for various excitation current parameters when the plasmonic metal is gold, are shown in Figure 4; the current amplitude $I_{\mathrm{surf}}$ is assumed constant within its range and equal to 1 A/m.

Under these conditions, Figure 4a illustrates the dependence of the SPP amplitude $E_{0z}$ on the spatial modulation period when the wavelength of SPP remains constant. The vacuum wavelength is set to be λ=0.8 μm, which yields the corresponding plasmon wavelength $\lambda_{\mathrm{SPP}}=0.783$ μm with its typical propagation length $l_{\mathrm{SPP}}=1/k_{I}=92$ μm. Figure 4a demonstrates that when the values of $\lambda_{\mathrm{SPP}}$ and Λ are inconsistent, the plasmon polariton amplitude oscillates as $z_{0}$ increases. These oscillations decay when $z_{0}\gg l_{\mathrm{SPP}}$, then the amplitude reaches the stationary value. If $\lambda_{\mathrm{SPP}}=\Lambda$, the stationary value is the largest and reached without oscillations.

The behavior of the curves can be interpreted as a result of excitation in the current zone of the “free” and “forced” SPP waves with the wave numbers $k_{\mathrm{SPP}}$ and $k_{g}$, respectively. These two waves have opposite-signed amplitudes, so at the beginning of the excitation current zone, at $z=-z_{0}$ (Figure 1b), the amplitude of the total right-propagating wave is zero. At the end of this zone, at z=0, the total wave is the sum of the beats of the “free” and “forced” waves, the first of which gradually decays due to the non-zero imaginary part of the plasmon polariton propagation constant $k_{\mathrm{SPP}}$. It is possible to show that, in this representation, the total field is proportional to $1-e^{-z_{0}k_{I}+i\Delta kz_{0}}$, i.e., as it is assumed in expression (Equation 6). It gives the initial amplitude for an SPP wave that is free from the external action excited outside the current range at z>0. If $\lambda_{\mathrm{SPP}}\neq \Lambda$, fluctuations in the amplitude of the excited SPP are consequences of the beats mentioned above. In resonance, there is no beating due to the equality of the propagation constants of “free” and “forced” SPP waves.

Figure 4b shows the results of calculations of the dependence of the SPP amplitude on the current modulation period. The current amplitude and plasmon parameters are the same as in the previous example. The curves obtained for $z_{0}<l_{\mathrm{SPP}}$ have pronounced traces of “free” and “forced” waves’ beats. As the length $z_{0}$ increases, the beats disappear, and the frequency response narrows and acquires the Lorentzian profile with a half-width equal to $\lambda_{\mathrm{SPP}}^{2}{\left(\pi l_{\mathrm{SPP}}\right)}^{-1}$.

To illustrate the case when the current modulation period can go beyond the 770–800 μm range, as assumed in Figure 4a,b, Figure 4c presents the results of calculations of the dependence of the SPP resonant amplitude on the sinusoidal current zone length $z_{0}$ over a broader range of Λ.

Figure 4 also contains the results of the numerical calculation of the SPP-wave amplitude excited by a sinusoidal longitudinal current on the metal surface, obtained by the FEFD method implemented in COMSOL Multiphysics. The mesh and simulation area’s size in the numerical model were chosen so as not to affect the simulation results under its minor variation. To extract the initial amplitude of the excited SPP wave from the entire electromagnetic field generated by the source current, we used the fact that sufficiently far from the excitation current, the longitudinal electric field at the interface decays exponentially along the *Z*-axis with a damping constant corresponding to $k_{\mathrm{SPP}}$, which indicates the dominant contribution of the SPP wave to the total field. Considering this, the initial amplitude of the SPP wave was taken from the extrapolation from the distant point to z=0 using the known exponential law of SPP attenuation. This extraction method provides the same results compared to the one used in the first subsection despite being based on a slightly different approach.

One can see that the numerical and analytical results agree well with each other. Therefore, we argue that the considered approach describes well the process of SPP generation by a surface current. Thus, the problem of plasmon polariton excitation due to the sinusoidal perturbation of the metal boundary (i.e., grating) reduces effectively to the appropriate choice of the current density. Of course, one can later introduce additional adjustments for better agreement with the experimental data since the current distribution may not be purely surface.

### 2.3. SPP Excitation by a Plane Wave Illumination

We are now ready to describe the SPP excitation with the approach developed in the previous section. We will start with the same corrugation as in Section 2.1, but introduce another approximation for E in (Equation 2), namely $E=\left(1+r\right)E_{0\_\mathrm{ins}}n_{z}$. Like in the previously investigated case, $E_{0\_\mathrm{ins}}$ is the incident wave field amplitude, and $r=\left(1-\sqrt{\varepsilon_{\mathrm{Me}}}\right)/\left(1+\sqrt{\varepsilon_{\mathrm{Me}}}\right)$ is the reflection coefficient from the flat metal surface. While the grating’s amplitude is chosen to be much less than the vacuum wavelength, it suggests to be a more accurate approximation.

Note that the current density $J_{g}$ will be zero everywhere except in the corrugated regions highlighted in Figure 1a in orange and blue, where it is $J_{g0}n_{z}$ and $-J_{g0}n_{z}$, respectively, and $J_{g0}=-ik_{0}\left(\varepsilon_{\mathrm{Me}}-1\right)\;\left(1+r\right)\rho_{V}^{-1}E_{0\_\mathrm{ins}}$. This gives the corresponding limits of integration over the variable *x* in (Equation 5), when substituted into it $J_{g}$ as the excitation current J. After performing the substitution, the expression for the SPP amplitude will include integrals of the form $\int_{-z_{0}}^{0}\exp\left(-\gamma_{V}x_{0}\cos^{+}\left(k_{g}z\right)\right)\;e^{-ik_{\mathrm{SPP}}z}\;dz$ and $\int_{-z_{0}}^{0}\exp\left(\gamma_{\mathrm{Me}}x_{0}\cos^{-}\left(k_{g}z\right)\right)\;e^{-ik_{\mathrm{SPP}}z}\;dz$, where the half-wave rectified cosine functions $\cos^{+}x$ and $\cos^{-}x$ are given by $\cos^{+}x=\left\{\begin{array}{cc} \cos x, & \cos x⩾\;0\\ 0, & \cos x<0 \end{array}\right.$ and $\cos^{-}x=\left\{\begin{array}{cc} 0, & \cos x⩾\;0\\ \cos x, & \cos x<0 \end{array}\right.$, so that the half-wave rectified cosine functions keep their sign for all *x*. The analytical calculation of these integrals becomes possible by decomposing exponents containing half-wave rectified cosine functions into a power series. If we limit the accuracy of such an expansion to the quadratic terms in $x_{0}$, after integration, the amplitude *Z* component of the SPP field turns out to be
(7)$$E_{0z}=\frac{i\gamma_{E}}{\Delta k+ik_{I}}\left(1-e^{-z_{0}k_{I}+i\Delta kz_{0}}\right)E_{0\_\mathrm{ins}},$$
where
(8)$$\gamma_{E}=\gamma_{0}\;\left(x_{0}-i\frac{2}{3\pi }k_{0}x_{0}^{2}\sqrt{\varepsilon_{\mathrm{Me}}+1}\right)$$
with
(9)$$\gamma_{0}=-i\left(\varepsilon_{\mathrm{Me}}-1\right)\left(1+r\right)\frac{k_{\mathrm{SPP}}^{2}}{2\varepsilon_{\mathrm{Me}}}\left|\frac{\varepsilon_{\mathrm{Me}}^{3/2}}{\varepsilon_{\mathrm{Me}}^{2}-1}\right|.$$

Let us add that we could use a less accurate way to account for the excitation current in the grating, which consists of first summing this current along the vertical axis (the *X* axis in Figure 1a) and then representing it in a purely surface form (Equation 5). Later, we will refer to this approach as the “flat current” approximation. It is not difficult to show that the amplitude of such a current will be $I_{\mathrm{surf}}=-ik_{0}\left(\varepsilon_{\mathrm{Me}}-1\right)\;\left(1+r\right)x_{0}E_{0\_\mathrm{ins}}/\rho_{V}$. Substituting it into (Equation 6) gives almost the same result as (Equation 7), except for the absence of a quadratic term in the parameter $x_{0}$ in (Equation 8). It is not difficult to understand that this term disappears in the flat current approximation because one does not consider the plasmonic mode’s finite depth of penetration into metal and air when integrating in (Equation 6). The power expansion of the exponents in the transverse SPP mode accounts for this feature. However, the necessity of introducing the terms of the third or higher degree of $x_{0}$ to improve the accuracy of the SPP amplitude calculation, as compared to the flat current approximation, is still obscure. We will address these questions when comparing the analytical and numerical calculations.

So far, we have not discussed the impact of radiation losses introduced by the scattering of the SPP wave on the grating. We will account for them in the following considerations. Let us differentiate expression (Equation 7) with respect to $z_{0}$. We obtain that $\frac{dE_{0z}}{dz_{0}}=-\gamma_{E}E_{0\_\mathrm{ins}}-\left(k_{I}-i\Delta k\right)\;E_{0z}$. The last expression is a differential equation for the plasmon amplitude, in which the energy dissipation is given by the terms $k_{I}\;E_{0z}$. It is reasonable to assume that the radiative losses, as well as thermal losses, are proportional to the SPP amplitude. Adding the corresponding term transforms the considered differential equation to the following form:(10)$$\frac{dE_{0z}}{dz_{0}}=-\gamma_{E}E_{0\_\mathrm{ins}}-\left(k_{I}+\gamma_{\mathrm{rad}}-i\Delta k\right)\;E_{0z}.$$

One should also make a corresponding substitution into (Equation 7):(11)$$E_{0z}=i\gamma_{E}\frac{1}{\Delta k+i(k_{I}+\gamma_{\mathrm{rad}})}\left(1-e^{-z_{0}\left(k_{I}+\gamma_{\mathrm{rad}}\right)+i\Delta kz_{0}}\right)E_{0\_\mathrm{ins}},$$
which, consequently, becomes an exact solution of Equation (Equation 10). Note that the introduction of radiation losses increases
(12)$$z_{\mathrm{sat}}={\left(k_{I}+\gamma_{\mathrm{rad}}\right)}^{-1},$$
which is the typical grating length at which the SPP amplitude reaches saturation, which now turns out to be smaller than $l_{\mathrm{SPP}}$. Hereafter, we will refer to the case where the grating length is much greater than $z_{\mathrm{sat}}$ as the long grating case. The value of $z_{\mathrm{sat}}$ also specifies the half-width of the resonance contour formed by the dependence of the SPP intensity on the period of the long grating, according to the expression
(13)$$FWHM=\lambda_{\mathrm{SPP}}^{2}{\left(\pi z_{\mathrm{sat}}\right)}^{-1}.$$

### 2.4. Evaluation of $\gamma_{\mathrm{rad}}$

Despite the fact that (Equation 11) takes proper account of the impact of losses on the SPP amplitude, the value of $\gamma_{\mathrm{rad}}$ remains unknown. Note that the differential Equation (Equation 11) turns out to be very similar to the one considered in [27], although the latter was obtained from different considerations. Using the approach proposed in the mentioned paper, let us add to the differential equation one more relation for the field intensity of the reflected light:(14)$$E_{\mathrm{ref}}=E_{r}+E_{\mathrm{lkg}},$$
where
(15)$$E_{r}=rE_{0\_\mathrm{ins}}$$
is the electric field intensity of the wave reflected directly from the flat metal surface, and
(16)$$E_{\mathrm{lkg}}=\kappa E_{0z}$$
is the electric field intensity of the wave arising as a result of plasmon energy leakage with κ as a coefficient characterizing this leakage.

When analyzing the resulting system of Equations (Equation 10) and (Equation 14), one should ensure that the grating length is sufficient to consider the incident and reflected waves as plane waves. In this case, they may not be separated into separate spatial Fourier components, as was done in [27]. Otherwise, the approach to solving the system of equations remains the same as in the above work. Using the transformations based on energy conservation, time-reversal, and geometry mirror symmetry [27], and the results for the lossless case ($k_{I}=0$), we find that the parameters $\gamma_{E}$, $\gamma_{\mathrm{rad}}$, and κ are connected by the following relation:(17)$$\gamma_{\mathrm{rad}}=\frac{{\left|\kappa \right|}^{2}}{2N_{1}}=\frac{N_{1}}{2}{\left|\gamma_{E}\right|}^{2},$$
where $N_{1}=\frac{1}{2k}\left|\left(\varepsilon_{\mathrm{Me}}+1\right)\frac{\varepsilon_{\mathrm{Me}}^{2}-1}{\varepsilon_{\mathrm{Me}}^{3/2}}\right|$. This relation differs from the similar one given in [27] mainly due to the presence of the normalization factor $N_{1}$, which appears due to the change in the approach to the description of the spatial frequency spectrum of radiation. Assuming that this relationship remains valid in the lossy case, we then obtain the expression for the electric field of reflected light $E_{\mathrm{ref}}=\rho E_{0\_\mathrm{ins}}$, where ρ is the light reflection coefficient from the grating, which, if $z_{0}>z_{\mathrm{sat}}$, is equal to
(18)$$\rho =e^{i\phi }\frac{i\Delta k+\gamma_{\mathrm{rad}}-k_{I}}{i\Delta k-\gamma_{\mathrm{rad}}-k_{I}},$$
where ϕ is the phase shift in the reflection from a flat metallic surface. Finally, the expression for the SPP amplitude is obtained by substituting the value of the parameter $\gamma_{\mathrm{rad}}$ found from the relation (Equation 17) into expression (Equation 11).

It is important to note that application of the “lossless” relation (Equation 17) to the system of Equations (Equation 10) and (Equation 14) in the lossy case is an approximation that is justified only at sufficiently small values of parameter $k_{I}$, and which, otherwise, can lead to errors in calculating the SPP amplitude, which will be discussed later. Hereafter, we will refer to this approximation as the “$k_{I}\to 0$ approximation”.

Note that within this approximation, provided the dissipative and radiative losses of SPP are equal, the output of the long resonant (Δk=0, $\Lambda =\lambda_{\mathrm{SPP}}$) grating achieves the maximum possible SPP amplitude $\left|E_{0z}^{\left(\mathrm{SUP}\right)}\right|$. The reflection coefficient in this case, as (Equation 18) suggests, becomes zero. The height of the grating profile, ensuring that $\gamma_{\mathrm{rad}}=k_{I}$, as follows from (Equation 8) and (Equation 17), can be calculated as
(19)$$x_{\mathrm{opt}}=\left(\sqrt{1+4bx_{1}}-1\right)/2b,$$
where $x_{1}=\sqrt{2k_{I}/N_{1}}/\left|\gamma_{0}\right|$, b=4Re(−iεMe+1)/3λ. The data obtained by calculating the values $x_{\mathrm{opt}}$ and $\left|E_{0z}^{\left(\mathrm{SUP}\right)}\right|$ for some wavelengths of the optical range are presented in Table 1.

The results of the calculations of the SPP amplitude dependence on the parameters of the gold grating, carried out following expressions (Equation 11) and (Equation 17) for the case of a normal incidence of a plane wave with λ=0.8 μm, are shown in Figure 5.

Figure 5 also illustrates the dependence of the SPP amplitude on the grating length $z_{0}$ when the height of its profile is $x_{0}=5$ nm. Note that the magnitude of the resonant grating period coincides with the SPP wavelength equal to $\lambda_{\mathrm{SPP}}=0.783$ μm. As one would expect, the curve corresponding to this period in Figure 5a exhibits monotonic growth up to the saturation value. The oscillations in non-resonant curves are similar to those in Figure 4a for the SPP current excitation case.

Figure 5b shows the dependence of the resonant ($\Lambda =\lambda_{\mathrm{SPP}}$) SPP amplitude $E_{0z}^{\left(\mathrm{rns}\right)}$ on the grating length $z_{0}$. Note that the saturation length $z_{\mathrm{sat}}$ for these dependencies is no longer constant. It is nearly constant as long as the grating profile height remains much smaller than $x_{\mathrm{opt}}=17.7$ nm (solid curves at $x_{0}$ = 1, 2, 4 nm). Otherwise, it decreases sharply due to an increase in radiation damping (solid curves at $x_{0}$ = 25 and 75 nm), as follows from expression (Equation 12).

Figure 5c shows the dependence of the amplitude $E_{0z}^{\left(\mathrm{sat}\right)}$ of the SPP excited by the long grating on its period. As expected, these curves have a Lorentzian profile similar to that for SPP excitation by a surface current. The resonance period, equal to 0.783 μm, is constant for all dependencies because the used model does not allow variation in the SPP wavelength as the height of the corrugation changes.

In Figure 5d, which shows the dependence of the amplitude $E_{0z}^{\left(\mathrm{sat}+\mathrm{rns}\right)}$ of the plasmon excited by the long resonant grating on the height of its profile (curve 1), one can see the presence of a maximum at $x_{0}=x_{\mathrm{opt}}$ with the highest amplitude value $\left|E_{0z}^{\left(\mathrm{SUP}\right)}\right|$.

The nature of the dependence of the SPP amplitude on the grating parameters does not change for other wavelengths of the optical range compared to the one in Figure 5; therefore, these dependencies are not given. Of course, the values $x_{\mathrm{opt}}$ and $\left|E_{0z}^{\left(\mathrm{SUP}\right)}\right|$ for other wavelengths are different, as supported by Table 1.

Although this paper is devoted to the problem of SPP excitation by the grating and not to the reflective properties of the latter, some intriguing features of SPP excitation, as we will see later, are also apparent in the analysis of the grating reflection coefficient. In this regard, Figure 6 presents the results of analytical calculations of the reflectance $R={\left|\rho \right|}^{2}$, depending on the parameters of the long grating obtained per expression (Equation 18) at λ=0.8 μm. In this case, the solid curves in Figure 6a illustrate the dependence of the coefficient *R* on the grating period, and Figure 6b shows the dependence of the resonance value of this coefficient on the height of the grating. As the amplitude $x_{0}$ increases, the presented dependencies show first an increase in the depth of the dip in the grating reflection, up to the case R=0, which occurs when the condition $x_{0}=x_{\mathrm{opt}}$ ($\gamma_{\mathrm{rad}}=k_{I}$) is satisfied. After that, the dip depth decreases, and the dependence expands due to the rapid growth of radiation attenuation.

The results of numerical calculations of the SPP amplitude at λ=0.8 μm performed for the geometry shown in Figure 1a, with the same technique of extracting the plasmon polariton contribution from the scattered field, as in the case of SPP wave excitation by a surface current, are presented by dotted and dashed curves in Figure 5. As can be seen, they agree fairly well with the analytical ones at sufficiently small (no more than 5 nm) values of the grating profile height. With a further increase in the profile height, one can observe a mismatch between the numerical and analytical results. In particular, in Figure 5c, the resonant wavelength shifts toward a shorter grating period, while for the analytical dependencies, the value of the resonant period is constant. This difference appears to be due to the effect of changes in the SPP mode propagation constant in the perturbed waveguide [24], the consideration of which is beyond the scope of this paper. However, we emphasize that we take the numerical values of the SPP resonance amplitude as the maximum values of the dashed dependencies in Figure 5c. Figure 5b shows that these values agree well with the analytical one, not only when $x_{0}\ll x_{\mathrm{opt}}$ (curves at $x_{0}$ = 1, 2, 4 nm), but also at much higher than $x_{\mathrm{opt}}$ heights (see the curve at $x_{0}$ = 75 nm). The differences, however, are mainly observed at $x_{0}\approx x_{\mathrm{opt}}$ (curve at $x_{0}$ = 25 nm). This is also clearly noticeable when comparing the numerical (curve 2) and analytical (curve 1) results of calculations of the dependence of the SPP resonance amplitude on the height of the long grating profile $x_{0}$ in Figure 5d. This figure shows that the magnitude of the discrepancy between the analytical and numerical results reaches approximately 20% at $x_{0}=x_{\mathrm{opt}}$ (for other $x_{0}$ up to $3x_{\mathrm{opt}}$, the discrepancy does not exceed 25%). Despite this, the positions of the maxima of the numerical and analytical curves visually coincide at $x_{\mathrm{opt}}\approx 18$ nm. The agreement between the analytical and numerical data at a high grating height continues, at least up to the value of 100 nm ($x_{0}\approx 6x_{\mathrm{opt}}$). The same picture is observed for other calculated wavelengths, as illustrated by the results given in Table 1.

It is also interesting to compare the widths of the analytical and numerical resonance dependencies presented in Figure 5c. When the height of the grating profile is low, these curves virtually coincide in width (curves at $x_{0}=4$ nm). However, as the height $x_{0}$ increases, the numerical resonance curves are narrower than the analytical ones. This feature is illustrated by curves 4 and 5 in Figure 5d, the first of which represents the results of analytical calculations of the FWHM value, according to (Equation 13), while the second is derived from numerical simulations.

Considering the reasons for the discrepancy between numerical and analytical results, we note that, in expression (Equation 8), if we reject the quadratic on the parameter $x_{0}$ term and pass to the “flat current” approximation, it leads to a much more pronounced difference between analytical curve 3, obtained in the approximation, and numerical dependence 2 in Figure 5d. Such a mismatch occurs not only at $x_{0}\approx x_{\mathrm{opt}}$, as it was when comparing curves 1 and 2, but also if $x_{0}>x_{\mathrm{opt}}$, when it can reach 100% or more. In addition, the maximum of curve 3 (“flat current” approximation) shifts to the right, and now its position is remarkably different from the numerical value of $x_{\mathrm{opt}}$. The additional quadratic term in (Equation 8) leads to a substantially better agreement between the corresponding analytic curve 1 and the numerical curve 2. Although one would expect even better agreement between the analytical and numerical results after introducing the cubic terms in (Equation 8), direct analytical calculations show that taking this term into account practically does not change the form of the analytical curve 1 in Figure 5d for heights in the range of 0–100 nm. Therefore, one should look for another source of discrepancy between the analytical and numerical calculations. Such a difference may be related to the limitations of the Born and $k_{I}\to 0$ approximations.

A comparison of the results of numerical calculations of the grating reflectance, which are presented as dashed curves, and in Figure 6 for the λ=0.8 μm case, with solid analytical curves, suggests the possible origins of this issue. The numerical results presented in the figure were obtained in the approximation of an infinitely long grating illuminated by an unbounded normal incident plane wave. Thanks to the periodicity, we performed all calculations within one grating period with periodic boundary conditions along the *Z* axis and a perfectly matched layer (PML) to absorb reflected light at the upper boundary of the calculation domain.

Figure 6 demonstrates that the numerical values of the *R* coefficient tend to the value of the reflectance from a flat gold surface, equal to 0.988, in two cases: at $x_{0}\to 0$ and away from resonance. The analytical value for *R* under the same conditions tends to 1. This difference is an obvious consequence of the $k_{I}\to 0$ approximation since this approximation ignores dissipative losses, without which a flat surface of any metal has R=1.

Another consequence of using this approximation seems to be more significant. A comparison of the analytical and numerical curves in Figure 6a exhibits a marked difference in the dip depth even at $x_{0}\ll x_{\mathrm{opt}}$ (the curve at $x_{0}=4$ nm in Figure 6a). At the same time, the results of numerical and analytical calculations of the SPP amplitude at this grating’s height presented in Figure 5 agree well with each other, which advocates the hypothesis that there is an error in the analytical calculation of the $\gamma_{\mathrm{rad}}$ value due to the $k_{I}\to 0$ approximation.

Indeed, it follows from the relation (Equation 11) that the amplitude of the SPP excited by the long resonant grating is limited by the sum of dissipative and radiative losses according to the expression $E_{0z}^{\left(\mathrm{sat}+\mathrm{rsn}\right)}=\gamma_{E}\frac{1}{k_{I}+\gamma_{\mathrm{rad}}}E_{0\_\mathrm{ins}}$. At $x_{0}\ll x_{\mathrm{opt}}$, the radiative losses are negligible compared to the dissipative losses. Therefore, the amplitude $E_{0z}^{\left(\mathrm{rsn}\right)}$ weakly depends on $\gamma_{\mathrm{rad}}$ and, thus, on the error of its calculation for small values of $x_{0}$. It is not the case, however, for the reflected wave. The first term in (Equation 14) for the reflected wave amplitude does not depend on the radiation losses, and the second term can be written as $\left|E_{\mathrm{lkg}}\right|=-\sqrt{2N_{1}\gamma_{\mathrm{rad}}}E_{0z}$, as in (Equation 17). Therefore, the inaccuracy of the calculation of the parameter $\gamma_{\mathrm{rad}}$ will directly affect the value of this summand and, as a consequence, the intensity of the reflected wave. Thus, the error in the calculation of the value of the coefficient *R* will appear regardless of the fulfillment of the condition $\gamma_{\mathrm{rad}}\ll k_{I}$, or equivalently, whether or not the grating’s height is relatively small.

Further discussing the source of the discrepancy between the analytical and numerical calculations, we note that if we fix the height of the grating profile at 4 nm but increase the value of $\gamma_{\mathrm{rad}}$ by a factor of 1.87, the corresponding analytical dependence for R(Λ) in Figure 6a (grey dashed curve) will almost completely coincide with the numerical one. With this modification of radiation attenuation, the analytical dependence $E_{0z}^{\left(\mathrm{rsn}\right)}\left(z_{0}\right)$ for the resonant plasmon amplitude will shift slightly down in Figure 5b and coincide completely with the dashed curve for $x_{0}=4$ nm (not shown in this figure). One might expect that for other grating heights, such a technique would help to eliminate the discrepancy between the analytical and numerical dependencies for the SPP amplitude in Figure 5b. However, the magnitude of this correction differs for different $x_{0}$. Thus, in the case where $x_{0}=x_{\mathrm{opt}}$, the coincidence of the analytical and numerical values of $E_{0z}^{\left(\mathrm{rsn}\right)}\left(z_{0}\right)$ is achieved when $\gamma_{\mathrm{rad}}$ is 1.52 times larger, and when $x_{0}>x_{\mathrm{opt}}$ no additional correction is required. One can interpret these features as follows. As long as $x_{0}\ll x_{\mathrm{opt}}$, dissipative SPP losses dominate and cannot be assumed to be small, thus reducing the accuracy of the $\gamma_{\mathrm{rad}}$ parameter calculation within the $k_{I}\to 0$ approximation and, as a result, requiring the maximum value of the correction factor. As the height of the grating profile increases, the contribution of dissipative losses to the SPP attenuation gradually decreases and eventually becomes negligibly small compared to radiation losses when $x_{0}\gg x_{\mathrm{opt}}$. In this case, the accuracy of the approximation $k_{I}\to 0$ increases, leading to the complete coincidence of analytical results with numerical ones.

Figure 6a shows that as the profile height continues to increase, the resonance position shifts toward lower values of Λ (just as it does for the SPP resonance amplitude in Figure 5c). Therefore, we take the minimum value of the numerical dependence R(Λ) as the numerical value of the resonant reflection coefficient $R_{\mathrm{rns}}$.

In Figure 6b, curve 2 shows the results of numerical calculations of $R_{\mathrm{rns}}$ as functions of the profile height $x_{0}$ of the long grating at λ=0.8 μm. The minimum of this curve is slightly shifted to the left, relative to the minimum of analytical dependence 1, and is observed at the grating profile height $x_{R=0}=14.5$ nm, which differs from $x_{\mathrm{opt}}=17.7$ nm. Note that the numerically calculated height $x_{R=0}$ for other wavelengths of the optical range also turn out to be smaller than $x_{\mathrm{opt}}$, which is illustrated by the data in the results of the calculations of this height in Table 1.

The limitations of the $k_{I}\to 0$ approximation can partially explain the difference in the position of the minimum of the analytical and numerical curves for $R_{\mathrm{rns}}$. Indeed, if one accepts the validity of the conclusion drawn above (that the maximum of the SPP resonance amplitude observed when the condition $x_{0}=\;x_{\mathrm{opt}}$ ($\gamma_{\mathrm{rad}}=k_{I}$) is satisfied), then the reflectance $R_{\mathrm{rns}}$ cannot be zero at any $k_{I}\neq 0$ for the real lossy metal. Indeed, it follows from expressions (Equation 14)–(Equation 16) that $\gamma_{\mathrm{rad}}=k_{I}$ provides $\left|E_{\mathrm{lkg}}\right|=\left|E_{0\_\mathrm{ins}}\right|$, $\left|E_{r}\right|=\left|rE_{0\_\mathrm{ins}}\right|$. Within the $k_{I}\to 0$ approximation, $\left|r\right|=1$, so the waves leaking out and reflected from the interface have equal amplitudes $\left|E_{\mathrm{lkg}}\right|=\left|E_{r}\right|$. As a result, the destructive interferences of $E_{\mathrm{lkg}}$ and $E_{r}$ waves nullify the reflection coefficient. However, due to the dissipative losses in a real metal, the value of $\left|r\right|$ is always slightly less than 1, so $\left|E_{\mathrm{lkg}}\right|>\left|E_{r}\right|$, and there is no complete mutual damping of the reflected and outgoing waves in the considered case. Therefore, the reflectance from the lossy metal becomes zero at a slightly lower grating height than $x_{\mathrm{opt}}$, when the amplitude of the outgoing wave $E_{\mathrm{lkg}}$ equals $E_{r}$, attenuated due to the reflection losses. The value $x_{R=0}$ calculated from these considerations at λ=0.8 μm is 17.5 nm, which is still noticeably (by 3 nm) greater than the numerical value $x_{R=0}$ for this wavelength. This mismatch can seemingly no longer be attributed to the $k_{I}\to 0$ approximation and its source is in the limitations of the Born approximation. Indeed, the accuracy of the latter should decrease in the presence of a strong SPP wave arising at $x_{0}\approx x_{\mathrm{opt}}$. Therefore, at $x_{0}\gg x_{\mathrm{opt}}$, when such a wave attenuates, the accuracy of the $k_{I}\to 0$ approximation increases (as noted above), as well as the accuracy of the Born approximation. As a consequence, the numerical and analytical dependencies for $R_{\mathrm{rns}}\left(x_{0}\right)$ at large values of the grating profile height converge to each other, as in Figure 6b (at least while this height remains smaller than $6x_{\mathrm{opt}}$).

Despite the difference in the dip depths, the widths of the analytical and numerical dependencies R(Λ) are the same for the small grating heights (curves for $x_{0}=4$ nm in Figure 6a). However, as $x_{0}$ increases, the numerical dependencies appear slightly wider than the analytical dependencies. This feature is illustrated by curves 6 and 4 in Figure 5d, the first of which represents the results of numerical calculations of the width of the resonance at half its depth (FWHD), and the second one represents the results of the corresponding analytical predictions, which, as expected, coincide with the same calculations for the half-width of analytical dependencies $E_{0z}^{\left(\mathrm{sat}\right)}\left(\Lambda \right)$. Surprisingly, the numerical curve 6 for FWHD is shifted in the opposite direction from analytical curve 4 compared to numerical curve 5 for FWHM. This bidirectional shift most likely cannot be explained by the limitations of the $k_{I}\to 0$ approximation due to the use of the Born approximation.

Thus, the analytical model considered in this paper describes generally well the qualitative dependence of the amplitude of the excited SPPs on the parameters of the sinusoidal corrugation on the plasmonic metal surface. The quantitative coincidence of the analytical and numerical calculation results is observed for small ($x_{0}\ll x_{\mathrm{opt}}$) and large ($x_{0}>x_{\mathrm{opt}}$) heights of the corrugation. However, for grating amplitudes in between, the numerical value of the resonance period of the grating turns out to be somewhat smaller than the analytical one, which requires further specification of the value of the SPP propagation constant for this case. Nevertheless, the model under consideration fairly accurately predicts the grating height $x_{\mathrm{opt}}$, at which it provides the maximal SPP amplitude. However, the value of the amplitude differs in the analytical and numerical calculations by less than 25%, which appears to be the result of the constraints of $k_{I}\to 0$ and Born approximations.

## 3. Conclusions

We presented an analytical description of SPP excitation by arbitrary configurations of nanostructured surfaces using a well-known method for calculating guided-mode amplitudes in waveguide theory. In this work, we focused on two cases: the case of a finite-length sinusoidal flat current and the case of a finite-length sinusoidal corrugation on a plasmonic metal surface under the normal incidence of a plane wave. For the first case, the described approach perfectly agrees with the numerical experiment. For the second case, our model correctly predicts the optimum amplitude of the corrugation to maximize the SPP excitation efficiency. We compared our analytical solution with numerical simulations and found excellent agreement between the two for non-resonant excitation and when the corrugation height was much smaller or larger than the optimum grating height. When the grating amplitude approaches the optimal value, the mismatch between the analytical and numerical values of the SPP amplitude was no more than 25%. We discussed the limitations of our analytical model that led to this mismatch. We believe that our approach is useful for modeling various phenomena associated with SPP excitation in metal nanostructures fabricated by laser processing or other methods.

## Figures and Tables

**Figure 1 nanomaterials-13-02091-f001:**
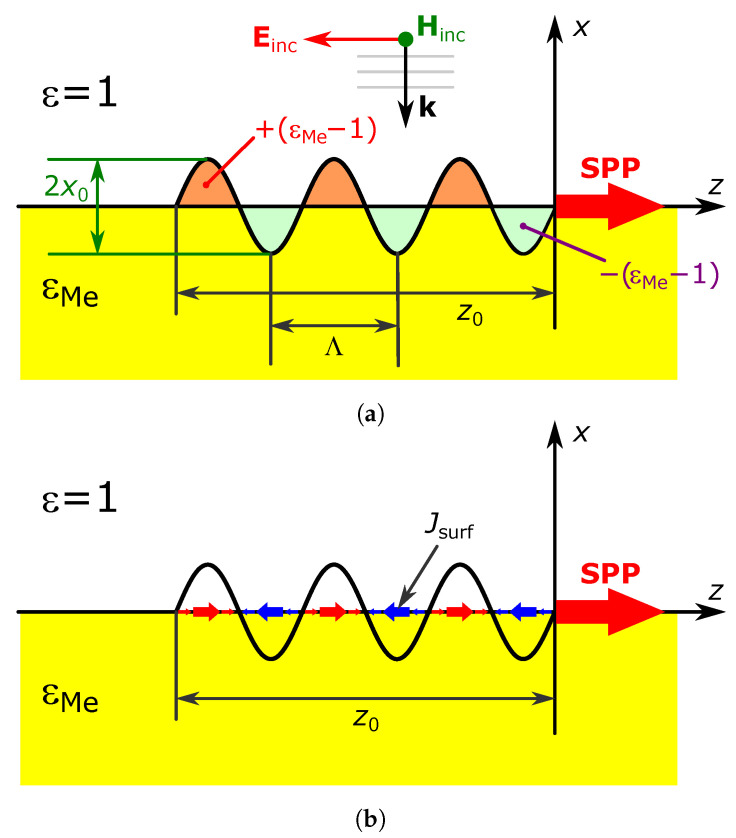
Geometry of surface plasmon polariton (SPP) excitation. (**a**) SPP excitation by a sinusoidally perturbed boundary. The amplitude of the forward-propagating plasmon polariton is calculated at point z=0, directly after the excitation zone. (**b**) Excitation of the SPP by a surface current.

**Figure 2 nanomaterials-13-02091-f002:**
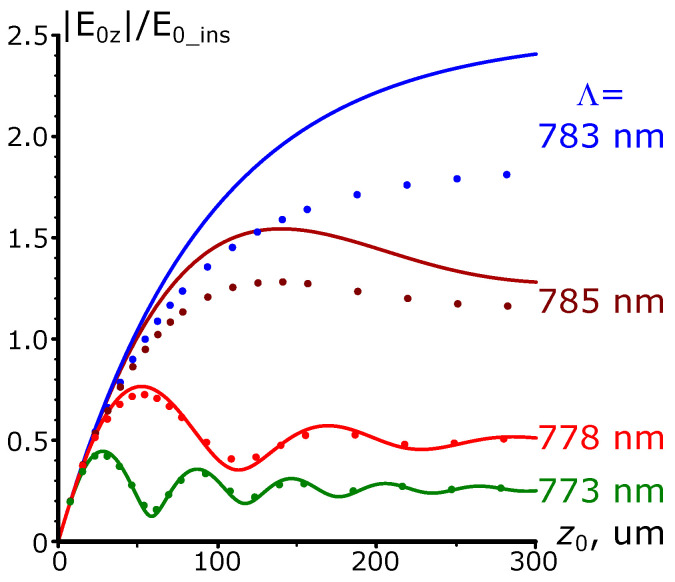
Comparison of the analytically calculated *Z* component of the electric field at the end of the gold grating and the results of the full-wave numerical simulations. The grating amplitude is 10 nm; the solid lines represent the analytical results while the dotted ones correspond to the numerical simulations; the vacuum wavelength is 800 nm.

**Figure 3 nanomaterials-13-02091-f003:**
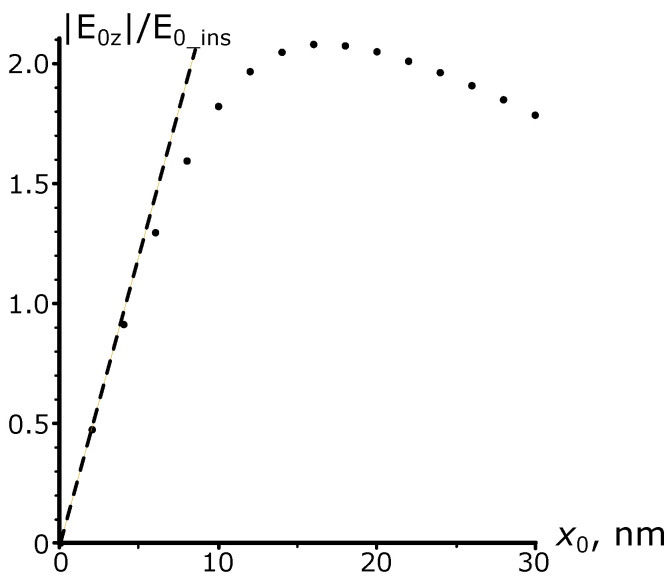
Dependence of the normalized *Z* component of the electric field near the end of the grating on the grating amplitude $x_{0}$ for the resonant SPP excitation, λ=800 nm. The dashed line shows the analytical dependence; the dots represent the numerical calculations. For both cases, the total length of the grating is kept constant and equal to 400.5 grating periods (≈313 μm).

**Figure 4 nanomaterials-13-02091-f004:**
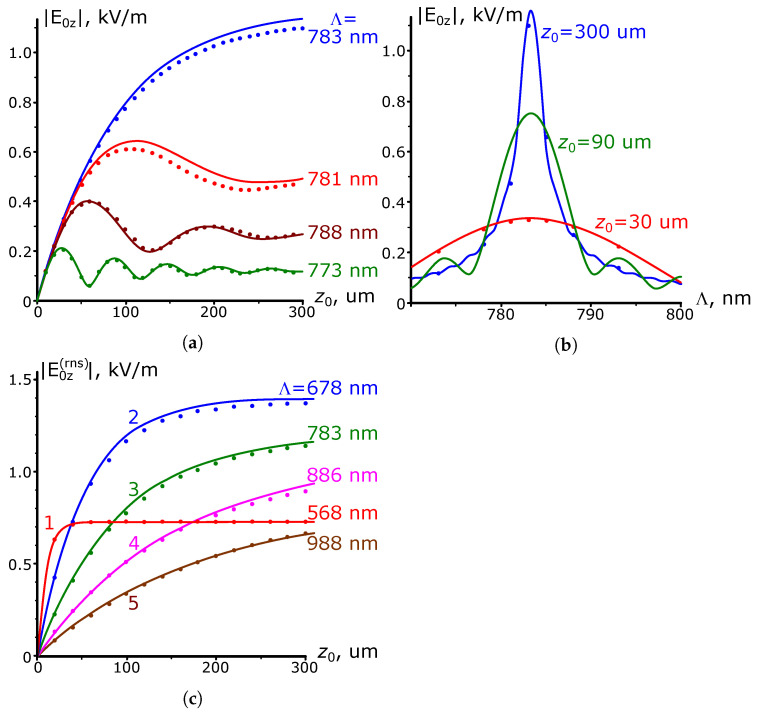
Dependence of plasmon amplitude on the excitation current parameters. The solid curves are the results of analytical calculations, the dotted ones are the results of numerical calculations. (**a**) Dependence of the SPP amplitude on the length of the sinusoidal current when the SPP parameters ($\lambda_{\mathrm{SPP}}=0.783$
μm, $l_{\mathrm{SPP}}=92$
μm, λ=0.8
μm) remain unchanged and the current modulation period Λ varies. (**b**) The dependence of the SPP amplitude ($\lambda_{\mathrm{SPP}}=0.783$
μm, $l_{\mathrm{SPP}}=92$
μm, λ=0.8
μm) on the current modulation period, with variation in the current zone’s length. (**c**) Dependence of the SPP resonance amplitude on the length of the sinusoidal surface current for different grating periods Λ. A vacuum wavelength λ is chosen to provide the resonance condition $\lambda_{\mathrm{SPP}}=\Lambda$. Curve 1: Λ=568 nm (λ=0.6
μm, $l_{\mathrm{SPP}}=9.6$
μm). Curve 2: Λ=687 nm (λ=0.7μm, $l_{\mathrm{SPP}}=52$
μm). Curve 3: Λ=783 nm (λ=0.8
μm, $l_{\mathrm{SPP}}=92$
μm). Curve 4: Λ=886 nm (λ=0.9
μm, $l_{\mathrm{SPP}}=147$
μm). Curve 5: Λ=988 nm (λ=1.0
μm, $l_{\mathrm{SPP}}=183$
μm).

**Figure 5 nanomaterials-13-02091-f005:**
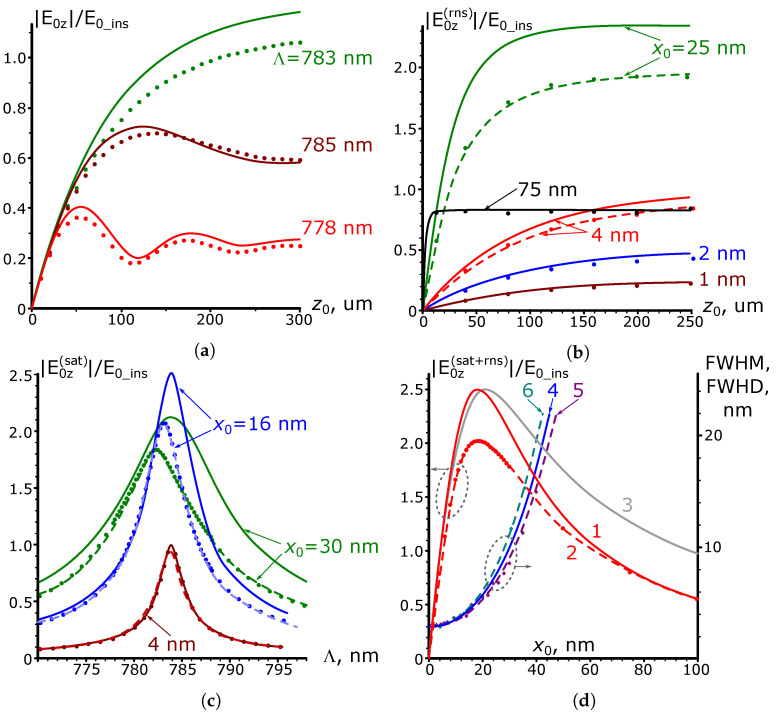
Dependence of SPP amplitude on the grating parameters. The solid curves are the results of analytical calculations, the colored dots and dashed lines are the numerical results. (**a**) Dependence of the SPP amplitude on the grating length $z_{0}$ with variation in the latter’s period. The height of the grating profile $x_{0}=5$ nm. The wavelength of radiation incident on the grating is λ=0.8
μm ($\lambda_{\mathrm{SPP}}=0.783$
μm). (**b**) Dependence of the SPP amplitude on the length $z_{0}$ in the resonant case ($\Lambda =\lambda_{\mathrm{SPP}}=0.783$
μm, λ=0.8
μm). (**c**) Dependence of the SPP amplitude on the grating period for long ($z_{0}\gg z_{\mathrm{sat}}$) gratings of different heights. (**d**) Dependence of the SPP parameters on the height $x_{0}$ of the long ($z_{0}\gg z_{\mathrm{sat}}$) grating ($\Lambda =\lambda_{\mathrm{SPP}}=0.783$
μm, λ=0.8
μm). Curves 1 and 3: analytical calculation of the SPP amplitude in the quadratic and linear (“flat”) surface current model, respectively; curve 2: numerical calculation of the SPP amplitude; curve 4: analytical calculation of full width at half maximum (FWHM) and full width of the resonance at half its depth (FWHD) for the SPP amplitude; curve 5: numerical calculation of the FWHM of SPP amplitude resonance dependencies; curve 6: numerical calculation of the FWHM of the grating resonant reflection dependencies.

**Figure 6 nanomaterials-13-02091-f006:**
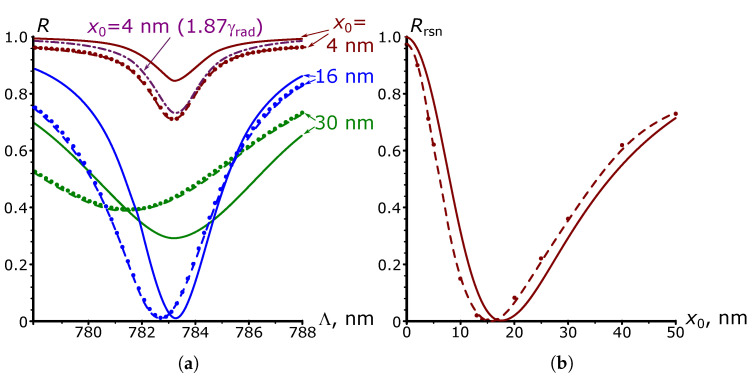
Analytical (solid curves) and numerical (dotted and dashed curves) dependencies of the reflectance on the parameters of a long grating in the case of a normal incidence of a plane light wave with λ=0.8
μm; (**a**) dependence of the reflection coefficient on the grating period; (**b**) dependence of the resonance reflection coefficient on the height of the grating profile.

**Table 1 nanomaterials-13-02091-t001:** Key parameters of the long gold grating and the amplitude of the excited SPP wave.

λ, μm	Analytical Calculations			Numerical Calculations		
	$x_{\mathrm{opt}}$, nm	$x_{R=0}$, nm	$|E_{0z}^{\left(\mathrm{SUP}\right)}|$	$x_{\mathrm{opt}}$, nm	$x_{R=0}$, nm	$|E_{0z}^{\left(\mathrm{SUP}\right)}|$
0.6	26.3	26.3	1.96$E_{0\_\mathrm{ins}}$	26	20	1.55$E_{0\_\mathrm{ins}}$
0.7	17.4	17.4	2.72$E_{0\_\mathrm{ins}}$	17	12	2.23$E_{0\_\mathrm{ins}}$
0.8	17.7	17.7	2.53$E_{0\_\mathrm{ins}}$	18	14.5	2.053$E_{0\_\mathrm{ins}}$
0.9	18.4	18.4	2.33$E_{0\_\mathrm{ins}}$	18	15	1.90$E_{0\_\mathrm{ins}}$

$x_{\mathrm{opt}}$ is the grating’s height of the profile where the SPP resonant amplitude reaches a maximum; $x_{R=0}$ is the height of the profile where the grating reflection coefficient is zero; $\left|E_{0z}^{\left(\mathrm{SUP}\right)}\right|$ is the maximum resonant amplitude of the longitudinal component of the SPP electric field at $z_{0}>z_{\mathrm{sat}}$ and $x_{0}=x_{\mathrm{opt}}$.

## Data Availability

Data sharing not applicable.

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
