# Peer review of "Scattering Amplitude of Surface Plasmon Polariton Excited by a Finite Grating"

_nanomaterials, 2023, doi:10.3390/nano13142091_

Round 1

Reviewer 1 Report

In this paper the authors present a series of models which seek to provide an analytical model for the electric field distribution due to the excitation of a surface plasmon polariton wave generated by a plane electromagnetic wave which is normally incident on a shallow metallic grating. The approach which is taken, which is based on the reciprocity theorem, is intuitively appealing and appears to agree well with numerical simulations. The authors use this model to make several predictions about optimal parameters for such gratings. To my knowledge this work is original and model such as this could be of great interest to the plasmonics community.

However, I have a number of concerns with the results that are presented, and (to a lesser extent) with the structure of the paper.

11.       The first approximate model for the electric field at the end of the grating is presented as equation (4). Firstly, in comparison to the equivalent expression given in the appendix (A11) it appears to be missing the imaginary term in the exponential component. However, more seriously it is not clear how this expression can generate the curves in Figure 2. The oscillatory term for equation (4) must come from the Exp[-i z0 kspp] (inserting the missing ‘i’) and so the period should be fixed by kspp. However, the curves in figure 2 show a range of periods, none of which are close to kspp. Can the authors please explain how Figure 2 was generated using Equation (4)? It is clearly very different from version in Equation (6) where there is an obvious source for the slow oscillation.

22.       I also tried to replicate the dashed line given in Figure 3 using equation (4). However, although obtained a straight line (as expected) the proportionality was very different (10^4 x larger). Similarly using Equation (6) I was able to partially reproduce Figure 4(a), but I could only achieve that by setting the value of x0 to 1000 x less than the value given in the text. I had the same experience with Figure 5(a) and equation (11). It is of course possible that I made a mistake but I suggest that the authors carefully check the expressions that are in the manuscript to make sure that they can reproduce all of the figures with the parameters that are given.

33.       Also related to this, how are the values of lspp obtained in part (c) of Figure 4? Using the permittivity data from Johnson and Christy and the relationship kI=Im[kspp] I obtain numerical values which are close, but which differ by at least 10%, which is troubling.

44.       In terms of the structure of the paper, information is not always presented in the most logical order. For example:

a.       Figure 1(b) is referred to before Figure 1(a)

b.       To attempt to reproduce Figure 2 we need to have expressions for reflectance r and the permittivity of the metal (epsilon_me) but these are not given until later.

c.       In figure 2 the analytical results are compared with numerical results, but the numerical approach is only described later in the text.

55.       Some other more minor issues:

a.       To avoid confusion, it would be helpful to refer to the vacuum wavenumber explicitly as k0 (assuming that’s what is represented by ‘k’).

b.       Equation (2) is stated as coming from Reference 22 (Snyder and Love) but my edition of Snyder and Love does not contain that equation. Please check if that is the correct source.

c.       On line 105 equation (2) is referred to as an integral, when it is not.

Author Response

Dear Reviewer,

We are pleased to receive many useful suggestions and questions from you.

In the attached file, you may find our answers to your questions and concerns as well as all the changes made by us to the revised version of the paper.

We hope that our common efforts improved the manuscript and are looking forward to assessing the changes made.

Reviewer 2 Report

In this manuscript, “Scattering amplitude of SPP excited by finite grating,” the authors present a method for calculating guided-mode amplitudes. Overall, this manuscript has a strong potential for another review round after applying the issues and addressing the shortcomings listed below:

1-The authors should polish/revise some grammatical mistakes and typos along the manuscript. I invite the authors to read their manuscript carefully and make the required changes where necessary.

2-Please increase the size of the text provided in the figures (where necessary).

3-Please increase the thickness of the lines within the figures (where necessary).

4-In the Introduction section, while discussing recent developments in the field of SPP excitation(s), the following works should also be considered and cited to give a more general view to the possible readers of the work: [(i) Monolithic Metal Dimer-on-Film Structure: New Plasmonic Properties Introduced by the Underlying Metal, Nano Letters 20, 2087-2093 (2020); (ii) Thermoplasmonics in Solar Energy Conversion: Materials, Nanostructured Designs, and Applications, Advanced Materials 2107351 (2022)].

5-In Figure 2, what is the reason for the discrepancy between blue solid and dotted lines? Please explain. Do the same in Figure 5a for the results shown in green.

6-For review purposes, please tell more about your COMSOL settings.

N/A.

Author Response

(The authors gave the same response as above.)

Round 2

Reviewer 1 Report

Thank you for making these revisions. The new version is much easier to follow.

Reviewer 2 Report

In its current form, the revised manuscript is good to proceed.